# Association between Adverse Events and Prognosis in Patients with Hepatocellular Carcinoma Treated with Atezolizumab Plus Bevacizumab: A Multicenter Retrospective Study

**DOI:** 10.3390/cancers14174284

**Published:** 2022-09-01

**Authors:** Shigeo Shimose, Hideki Iwamoto, Masatoshi Tanaka, Takashi Niizeki, Masahiko Kajiwara, Satoshi Itano, Etsuko Moriyama, Tomotake Shirono, Yu Noda, Naoki Kamachi, Masahito Nakano, Ryoko Kuromatsu, Hironori Koga, Takumi Kawaguchi

**Affiliations:** 1Division of Gastroenterology, Department of Medicine, Kurume University School of Medicine, Kurume 830-0011, Japan; 2Iwamoto Internal Medical Clinic, Kitakyusyu 802-0832, Japan; 3Clinical Research Center, Yokokura Hospital, Miyama 839-0295, Japan; 4Department of Gastroenterology, Chikugo City Hospital, Chikugo 833-0041, Japan; 5Department of Gastroenterology, Kurume Central Hospital, Kurume 830-0001, Japan

**Keywords:** adverse events, overall survival, hepatocellular carcinoma, overall survival, progression-free survival

## Abstract

**Simple Summary:**

This study aimed to evaluate the correlation between adverse events (AEs) and overall survival (OS) in patients with unresectable hepatocellular carcinoma (HCC) treated with atezolizumab plus bevacizumab (atezo/beva). Liver injuries were significantly correlated with shorter survival. In a logistic regression analysis, fatigue ≥ grade 2, liver injury ≥ grade 3, and modified albumin–bilirubin grade 2b were identified as independent factors for discontinuation due to AEs. We concluded that the development of liver injury was a negative factor for OS and that we should be vigilant in monitoring AE during atezo/beva treatments.

**Abstract:**

This study aimed to evaluate the correlation between adverse events (AEs) and overall survival (OS) in patients with unresectable hepatocellular carcinoma treated with atezolizumab plus bevacizumab (atezo/beva). This was a multicenter study in which 130 patients were enrolled. Hypertension and skin disorders had a significant correlation with longer survival (median survival time (MST): not reached vs. 14.3 months and not reached vs. 14.8 months, *p* = 0.001 and *p* = 0.047, respectively). In contrast, liver injuries were significantly correlated with shorter survival (MST: 14.7 months vs. not reached, *p* = 0.036), and the median development time was 21 days. In a logistic regression analysis, fatigue ≥ grade 2, liver injury ≥ grade 3, and modified albumin–bilirubin grade 2b were identified as independent factors for discontinuation due to AEs. The OS in the no discontinuation due to AE group was significantly longer than that in the discontinuation due to AEs group (MST not reached vs. 11.2 months, *p* = 0.001). We concluded that the development of liver injury was a negative factor for OS and that we should be vigilant in monitoring AE during atezo/beva treatments.

## 1. Introduction

Hepatocellular carcinoma (HCC) is the most common primary liver cancer and a leading cause of cancer-related deaths worldwide [1]. The reasons for these phenomena are that HCC is still often detected in the advanced stage [2] and that sorafenib (SORA) was the only approved systemic treatment [3] until a few years ago. However, a systemic treatment for HCC developed remarkably in the last few years, and various molecular targeted agents (MTAs) that mainly target tumor angiogenesis have recently been approved [4]. More recently, combination therapies with immune checkpoint inhibitors (ICI) and anti-angiogenic drugs have been approved. In 2020, the combination of atezolizumab, which targets programmed death-ligand 1, with bevacizumab, an antiangiogenic agent that targets vascular endothelial growth factor (atezo/beva), was established as first-line systemic chemotherapy for unresectable HCC [5]. According to the IMbrave 150 trial, this combination therapy was superior to SORA in terms of progression-free survival (PFS) and overall survival (OS). In real-world practice, several reports have shown that atezo/beva is a favorable therapeutic response and may have less influence on hepatic reserve functions than conventional forms of treatment [6,7,8]. Regarding adverse events (AEs), most patients treated with atezo/beva developed AEs, as reported in the updated data of the IMbrave 150 trial [9], and the development of severe AEs was associated with treatment discontinuation. Therefore, healthcare providers need sufficient information on the key aspects of safety profiles of such events. We previously reported that AEs are potential predictive factors for OS, and careful management to avoid the discontinuation of treatment can result in longer survival periods in patients receiving lenvatinib (LEN) [10,11]. However, it is still unclear whether the occurrence of AEs due to atezo/beva correlates with the prognosis of patients with HCC in real-world practice. This study aimed to investigate the correlation between AEs factors and the prognosis of patients with unresectable HCC treated with atezo/beva. Furthermore, we also investigated the profiles associated with discontinuation due to AEs.

## 2. Materials and Methods

### 2.1. Study Design and Patients

This study retrospectively evaluated 155 patients with unresectable HCC who were treated with atezolizumab plus bevacizumab between November 2020 and April 2022 at five institutions: the Kurume University Hospital (Kurume, Japan), Yokokura Hospital (Miyama, Japan), Iwamoto Internal Medicine Clinic (Kitakyushu, Japan), Kurume Central Hospital (Kurume, Japan), and Chikugo City Hospital (Chikugo, Japan). The data cut-off for this analysis was 30 June 2022. Of the included patients, 25 were excluded. In total, 130 patients enrolled in this study (Appendix A). The study was conducted in accordance with the Declaration of Helsinki and was approved by the ethical committee of the Kurume University School of Medicine (approval number: 20183). Informed consent was obtained using an opt-out approach.

### 2.2. Inclusion and Exclusion Criteria

The patient inclusion criteria for the present study were as follows: (1) diagnosis of HCC, (2) age > 18 years, (3) Eastern Cooperative Oncology Group performance status (PS) 0 or 1, and (4) complete follow-up until death or study cessation (30 June 2022). The patient exclusion criteria were as follows: (1) Child-Pugh class B or C, (2) PS > 1, (3) active esophageal varices, and (4) a history of autoimmune disease.

### 2.3. Treatment Protocol

Patients received 1200 mg of atezolizumab plus 15 mg/kg of bevacizumab intravenously every 3 weeks, according to pharmaceutical recommendations. The patients received treatment until the development of unacceptable AEs or tumor progression. Treatment was discontinued if any unacceptable or severe adverse event was observed.

### 2.4. Evaluation of the Therapeutic Response

The therapeutic response was evaluated using dynamic computed tomography or magnetic resonance imaging 3 weeks after the initiation of treatment according to the Response Evaluation Criteria in Solid Tumors version 1.1 (RECIST v1.1) [12]. This response was re-evaluated every 3 weeks until death or study cessation.

### 2.5. Assessment of Safety and Liver Function

AEs were assessed according to the Common Terminology Criteria for Adverse Events version 5.0 [13]. Treatment was continued until the appearance of unacceptable AEs or progressive disease. The albumin–bilirubin (ALBI) [14] score was examined at baseline and at 3, 6, 12, 18, and 24 weeks to assess changes in liver function.

### 2.6. Statistical Analysis

All statistical analyses were performed using the JMP statistical analysis software version 15 (JMP Pro version 15, Tokyo, Japan), and all data are presented as numbers or median (range). Continuous variables were compared using a one-way analysis of variance with Scheffe’s post hoc test. PFS and OS were calculated using the Kaplan–Meier method and analyzed using the log-rank test. Multivariate analyses were conducted using the Cox proportional hazards model to identify risk factors associated with OS. Statistical significance was defined as a two-tailed *p*-value < 0.05. We also performed a decision tree analysis to identify factors associated with discontinuations due to AEs, as previously described [15]. To select factors for multivariate analyses, a stepwise procedure was performed, as previously described [16].

## 3. Results

### 3.1. Patient Characteristics

The characteristics of the 130 enrolled patients are shown in Table 1. The median age was 72.5 years old, and 78.5% of patients were male. The median body mass index (BMI) was 23.1 kg/m^2^ (15.4–35.2 kg/m^2^). There were 40 (30.8%) patients with modified-ALBI (m-ALBI) grade 1, 45 (34.6%) patients with m-ALBI grade 2a, and 45 (34.6%) patients with m-ALBI grade 2b. The median aspartate transaminase and alanine aminotransferase levels were 41 and 27 mg/dL, respectively. The median tumor size was 33.0 mm, and Barcelona Clinic Liver Cancer stage C was observed in 46.9% of patients (61/130). The median follow-up time was 10.1 (1.4–20.5) months.

### 3.2. Initial and Best Therapeutic Outcomes of Atezo/Beva

The distribution of therapeutic responses to atezo/beva is presented in Table 2. In the initial RECIST evaluation, a complete response (CR) was observed in none of the patients, a partial response (PR) was observed in 35 patients (26.9%), stable disease (SD) was observed in 78 patients (60.0%), and progressive disease (PD) was observed in 17 patients (13.1%). The overall objective response rate (ORR) and disease control rates (DCR) were 26.9% and 86.9%, respectively. However, the overall ORR and DCR were 32.3% and 86.9%, respectively, for the best response.

### 3.3. PFS and OS Associated with Atezo/Beva

The median PFS was 6.4 months (Figure 1A), whereas the median survival time (MST) was 18.2 months (Figure 1B).

### 3.4. Changes in the ALBI Score during Atezo/Beva Treatment

Figure 2 shows the changes in the ALBI score at 3, 6, 12, 18, and 24 weeks from baseline after atezo/beva treatment. The median ALBI scores at 3, 6, 12, 18, and 24 weeks after introducing atezo/beva were −2.41, −2.22, −2.31, −2.27, −2.33, and −2.29, respectively. Although the deterioration in the ALBI score was significant at 3 weeks (−2.41 vs. −2.22, *p* =  0.011), there was no deterioration in the ALBI score in the subsequent treatment.

### 3.5. Adverse Events Profiles and Timing of AEs with Atezo/Beva

The AEs observed during the atezo/beva treatment are shown in Table 3. The overall incidence rate of any grade of AE was 96.9%, and the incidence rate of AE grade ≥ 3 was 36.1%. Among the patients included, 57 (43.8%) experienced liver injury, 54 (41.5%) experienced hypertension, 37 (28.5%) experienced proteinuria, 36 (27.6%) experienced fatigue, 32 (24.6%) experienced skin disorders, 30 (23.0%) experienced fever, 20 (15.3%) experienced appetite loss, and 17 (13.0%) experienced hemorrhage. The content of the grade ≥ 3 AEs included proteinuria (9.2%), hypertension (6.9%), hemorrhage (6.9%), and liver injury (6.1%). The timing of each AE from the start of atezo/beva is shown in Figure 3. The earliest AE was fever, with a median period of 14 (5–126) days. Median timings for liver injury, fatigue, appetite loss, proteinuria, diarrhea, hypertension, and skin disorders were 21 (2–217), 38 (6–174), 41 (9–190), 42 (7–169), 43 (9–236), 46 (2–407), and 52 (4–272) days, respectively.

### 3.6. Survival Analysis According to Each AE Profile following Atezo/Beva Treatment

The survival curves for each AE profile are shown in Figure 4 and Appendix A. Patients who developed liver injury had a significantly shorter survival time than those who did not (MST:14.7 months vs. not reached, *p* = 0.036) (Figure 4A). In contrast, patients who developed hypertension or skin disorder survived significantly longer than those who did not (MST: not reached vs. 14.3 months and not reached vs. 14.8 months, *p* = 0.001 and *p* = 0.047, respectively) (Figure 4B,C). Proteinuria, fatigue, appetite loss, and fever did not correlate with survival (Appendix A).

### 3.7. Survival Analysis According to Early and Late Onset AEs

We defined the early-onset of AEs that develop within 6 weeks and the late-onset of AEs that develop after 6 weeks. There were no differences in OS between the two groups (early-onset AEs:17.9 months vs. late-onset AEs: not reached, *p* = 0.658) (Appendix A).

### 3.8. Univariate and Multivariate Analyses of Factors Associated with OS

The ALBI grade, liver injury, hypertension, skin disorders, and post-progression treatments were identified as independent factors for OS in multivariate analysis (Table 4).

### 3.9. Changes in ALBI Score in Patients with or without Developed Liver Injury

The median ALBI score in patients without liver injury recovered to nearly baseline values at 6 weeks after the introduction of atezo/beva (−2.42 vs. −2.39, *p* = 0.29). However, the median ALBI score in patients with liver injury did not improve compared with that at baseline (−2.39 vs. −2.17, *p* = 0.03) (Figure 5).

### 3.10. Decision-Tree Analysis for the Discontinuation of Atezo/Beva Due to AEs

In this study, the rate of discontinuation due to AEs in all subjects was 32.2% at the time of study cessation (Appendix A). To determine the profiles associated with discontinuation owing to AEs, a decision tree analysis was performed. Fatigue was identified as the first splitting variable for the rate of discontinuation due to AEs. Although the rate of discontinuation due to AEs was only 26.7% in patients with fatigue <grade 2, the rate of discontinuation due to AEs was 80% in patients with fatigue ≥grade 2. In patients with fatigue <grade 2, the second splitting variable was liver injury. In patients with liver injury <grade 2 and liver injury ≥grade 3, the rates of discontinuation due to AEs were 20.5 and 87.5%, respectively. In patients with liver injury <grade 2, the ALBI grade was identified as the next level of splitting variables. In patients with fatigue <grade 2 concomitant with liver injury <grade 2 and ALBI grade 1 or 2a, the discontinuation rate due to severe AEs was only 12.8% (Figure 6).

### 3.11. Logistic Regression Analysis for Discontinuation Due to AEs

Fatigue ≥ grade 2, liver injury ≥ grade 3, and m-ALBI grade 2b were selected as variables in a stepwise logistic regression analysis. In the logistic regression analysis, all three variables were identified as independent factors for discontinuation due to AEs (Table 5).

### 3.12. Additional Treatments after the Discontinuation of Atezo/Beva

Until the time of study cessation, 96 (73.8%) patients had their atezo/beva treatment discontinued, and 68 (70.8%) patients received subsequent treatments (Table 6). Among these patients, 24 patients (35.2%) were treated with LEN, and 9 patients (13.2%) were treated with ramucirumab or transcatheter arterial chemoembolization.

## 4. Discussion

This study showed that liver injury, hypertension, and skin disorders were important factors in predicting the survival of patients with HCC treated with atezo/beva. Moreover, we demonstrated that ≥grade 2 fatigue, ≥grade 3 liver injury, and m-ALBI grade 2b were independently associated with discontinuation due to AEs in patients with HCC treated with atezo/beva.

This study demonstrated that the ORR and DCR in the included patients were 32.3% and 86.9%, respectively. Furthermore, incidence rates of any grade of and grade ≥3 AEs were 96.9% and 36.1%, respectively, in patients with HCC treated with atezo/beva. According to the IMbrave 150 trial, the ORR, DCR, and rate of grade ≥3 AEs were 30.0%, 74.0%, and 43.0% in patients with unresectable HCC treated with atezo/beva, respectively [9]. Thus, the therapeutic effect and incidence rate of AEs in the present study appear to be similar to those in previous reports, suggesting that the enrolled subjects, treatment effects, and use of atezo/beva in our study are standard.

In this study, patients who developed liver injury during atezo/beva treatments had a significantly shorter survival time than those who did not. Previous reports have shown that liver injuries are associated with poor prognosis in patients with cancer treated with ICI therapy [17,18,19]. Interestingly, the prognosis of patients with HCC with grade 1 or 2 liver injury during ICI treatments was also poor [20]. Chen et al. reported that patients with liver injury in liver cancers have the highest chance of concomitant hyperbilirubinemia and biliary obstruction as cholestasis may contribute to the development of liver injury [17]. In this study, the median ALBI score in patients with liver injury did not improve compared to that at baseline (−2.39 vs. −2.17, *p* = 0.03). Moreover, Mouri et al. reported that patients administered with systemic steroids had fewer therapeutic effects than those who did not [21]. In fact, 14% (8/57) of patients who developed liver injury received systemic steroid treatments in our study. This may have contributed to the poor prognosis associated with the appearance of liver injury throughout the atezo/beva treatment. In contrast, hypertension and skin disorders were better prognostic factors for survival. Several studies reported that hypertension is a clinical biomarker for responses to bevacizumab [22,23]. Moreover, we previously reported that the appearance of hypertension can be a clinically promising early surrogate marker for predicting the survival of patients treated with LEN [11]. Furthermore, one of the reasons is that patients who developed hypertension had a significantly higher DCR than those who did not in this study (Appendix A). As for skin disorders, the development of immune-related skin disorders was found to be correlated with favorable outcomes and its practical feasibility as a potential predictive surrogate marker [24,25]. The reason why hypertension and skin disorders could be positive prognostic markers is unclear. However, to our knowledge, this study is the first to reveal that hypertension and skin disorders could be positive predictive markers for the prognosis of patients undergoing atezo/beva treatment. Thus, further studies are needed to clarify these reasons.

Recently, a previous study reported that OS correlated with the treatment duration of systemic therapy [26]. In this study, we found that treatment durations were significantly longer in patients who developed hypertension and skin disorders than in patients who did not (median treatment duration: 7.8 vs. 4.9 months and 8.5 vs. 5.3 months, *p* = 0.001 and *p* = 0.027, respectively). These AEs are relatively manageable using supportive drugs such as antihypertensive drugs and steroids. Therefore, as possible, it would be better to continue the treatment with the management for these AEs. However, if the effects of treatment are no longer confirmed, we should promptly switch to second-line treatments. Moreover, the treatment duration was significantly shorter in patients who developed liver injury than in patients who did not (median treatment duration: 4.3 vs. 6.7 months, *p* = 0.034). Additionally, the median ALBI score in patients with liver in-jury did not improve compared with that at baseline, which suggests that the development of liver injury induces the deterioration of liver function. Thus, AE management is needed for patients who developed liver injury as possible, but switching to second-line treatments should be also considered at the time of the development of liver injury, particularly severe liver injury

In this study, fatigue ≥ grade 2, liver injury ≥ grade 3, and m-ALBI grade 2b were identified as independent factors for discontinuation due to AEs using multivariate analyses. Moreover, we also found that fatigue was the initial splitting variable for the rate of discontinuation due to AEs in patients with HCC treated with atezo/beva, followed by liver injury and m-ALBI grade in the decision tree analysis. Several previous studies have reported that fatigue, liver injury, and m-ALBI grade are predictive factors for therapeutic effects or OS in patients treated with systemic therapy [18,27,28]. Fatigue can be detrimental to the patients’ quality of life [29], and preserved liver function is a favorable factor related to eligibility for post-treatment [30]. Sequential systemic therapy has recently been considered an effective strategy for the treatment of unresectable HCC. Furthermore, discontinuation due to AEs reduces the rate of transition to the next treatment and must be avoided. In our study, among the discontinuation of atezo/beva treatment, 68 patients (70.8%) received subsequent treatment. However, patients who had their treatment discontinued due to Aes that were significantly less likely to receive subsequent treatments than those who did not (38.7% vs. 86.1%, *p* < 0.001). Moreover, the median OS in the not discontinued due to AEs group was significantly longer than in the discontinued due to AEs group (MST: not reached vs. 11.2 months, *p* = 0.001, Appendix A), indicating that our results were in accordance with those of previous reports treated with systemic therapy [27,28,31]. Thus, clinicians should be vigilant in monitoring AE during atezo/beva treatment. Moreover, the establishment of a comprehensive grading system to predict discontinuation due to AEs is needed in the management of atezo/beva treatment.

This study has some limitations. First, this was a retrospective study. Second, although atezo/beva treatment was administered after the disappearance of the effect of previous treatment, we cannot deny that previous MTAs influenced the development of AEs due to atezo/beva. Third, we investigated post-treatment after atezo/beva treatment and did not obtain a sufficient observation period. Thus, further prospective validation studies with long-term follow-up are required.

## 5. Conclusions

In this study, we showed that fatigue, liver injury, and m-ALBI grade 2b were independently associated with discontinuation due to AEs in patients with HCC treated with atezo/beva. The study revealed that the types of developed AEs were important in predicting survival in atezo/beva treatments. The establishment of appropriate AE management is needed to further contribute to survival in patients with HCC.

## Figures and Tables

**Figure 1 cancers-14-04284-f001:**
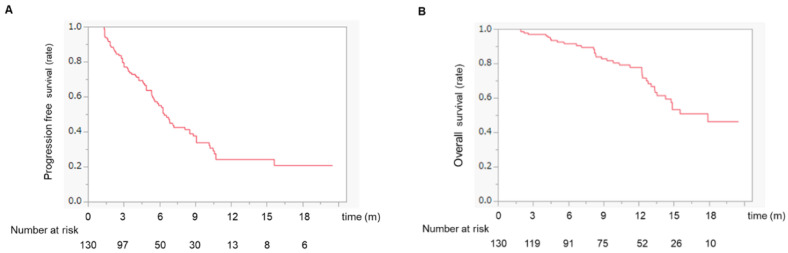
(**A**) Progression-free survival in HCC patients treated with atezo/beva. The median progression-free survival was 6.4 months. (**B**) Overall survival in HCC patients treated with atezo/beva. The median survival time was 18.2 months. Abbreviation: HCC, hepatocellular carcinoma; atezo/beva, atezolizumab plus bevacizumab.

**Figure 2 cancers-14-04284-f002:**
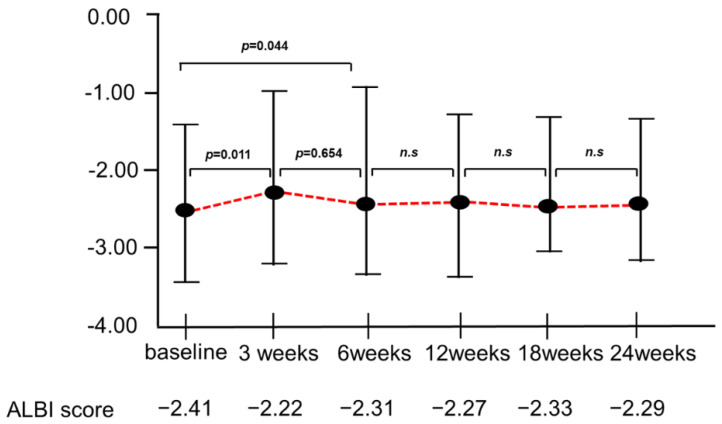
Changes in the ALBI score at 3, 6, 12, 18, and 24 weeks from baseline in the periods of atezo/beva treatments. Deterioration of ALBI score was significant at 3 weeks (−2.41 vs. −2.22, *p* = 0.011); however, there was no deterioration in the ALBI score in the subsequent treatment. ALBI, albumin–bilirubin; atezo/beva, atezolizumab plus bevacizumab; n.s, not significant

**Figure 3 cancers-14-04284-f003:**
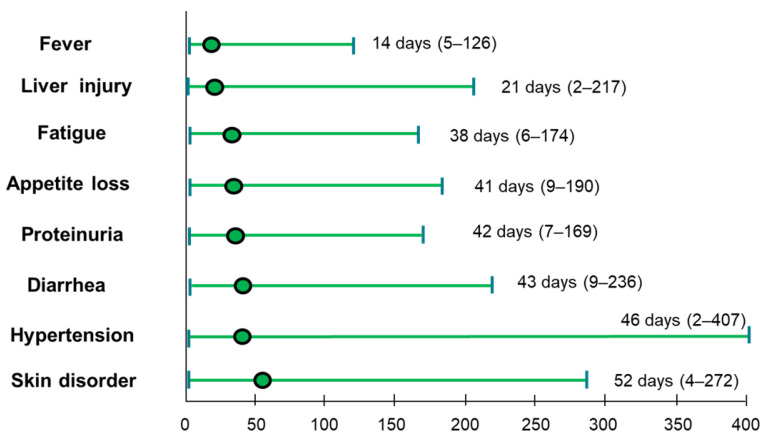
Time from the initial administration of atezo/beva to occurrence of AEs. The green dot indicates median time. AE, adverse events; atezo/beva, atezolizumab plus bevacizumab.

**Figure 4 cancers-14-04284-f004:**
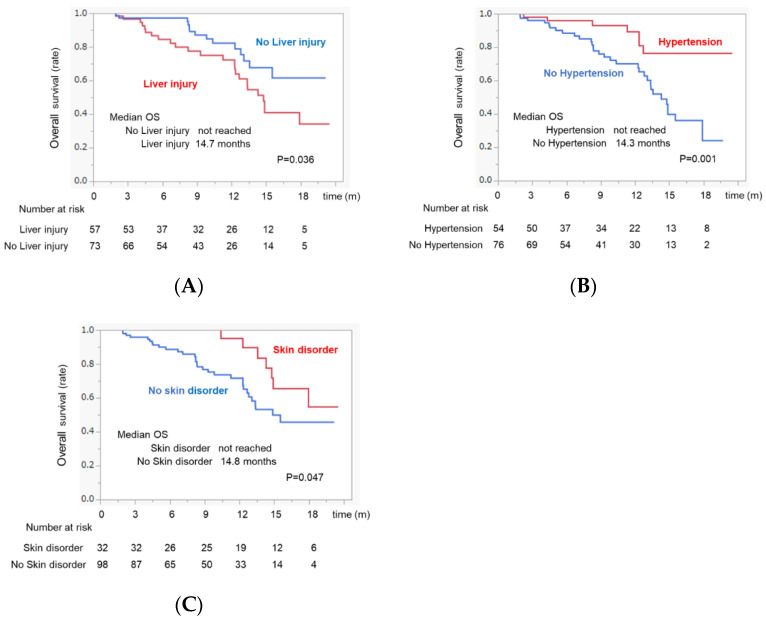
Overall survival time in patients who developed each AE in atezo/beva treatments. (**A**) Kaplan–Meier curves for overall survival according to whether a liver disorder is present. The blue line indicates the liver injury group. The red line indicates the no liver injury group. (**B**) Kaplan–Meier curves for overall survival according to whether there is hypertension. The red line indicates the hypertension group. The blue line indicates the no hypertension group. (**C**) Kaplan–Meier curves for overall survival according to whether there are skin disorders. The red line indicates the skin disorder group. The blue dotted line indicates the no skin disorder group. AE, adverse events; atezo/beva, atezolizumab plus bevacizumab.

**Figure 5 cancers-14-04284-f005:**
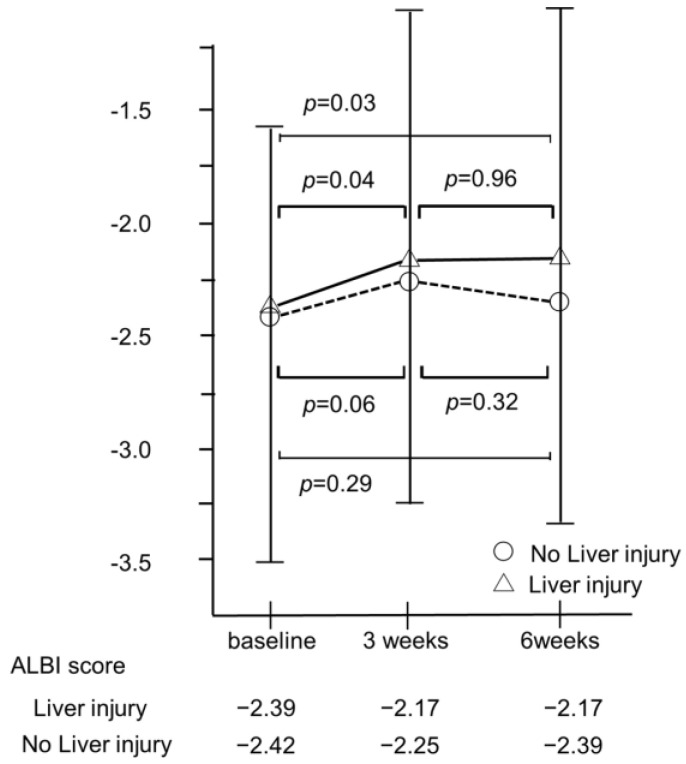
Changes in ALBI score 3 and 6 weeks after the atezo/beva treatment according to whether there is liver injury. The median ALBI score in patients without liver injury recovered to nearly baseline values at 6 weeks after introducing atezo/beva (−2.42 vs. −2.39, *p* = 0.29). However, the median ALBI score in patients with liver disorder has not improved compared to baseline values (−2.39 vs. −2.17, *p* = 0.03). HCC, hepatocellular carcinoma; ALBI, albumin–bilirubin; atezo/beva, atezolizumab plus bevacizumab.

**Figure 6 cancers-14-04284-f006:**
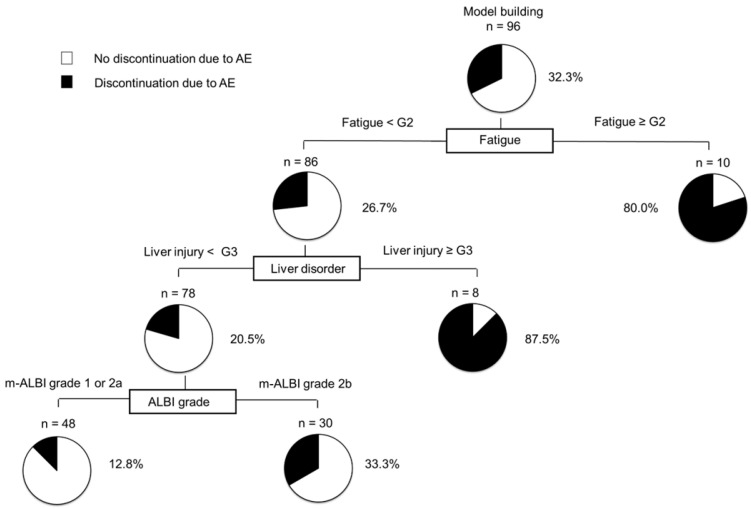
Profiles associated with discontinuation due to AEs in patients with HCC treated with atezo/beva. Decision-tree algorithm for discontinuation due to AEs. The pie graphs indicate the percentage of no discontinuation due to AEs (white)/discontinuation due to AEs (black) in each group. HCC, hepatocellular carcinoma; atezo/beva, atezolizumab plus bevacizumab.

**Table 1 cancers-14-04284-t001:** Patient characteristics.

Characteristic	All Patients
N	130
Age (years old)	72.5 (37–93)
Sex (female/male)	28/102
PS (0/1)	102/28
Body Mass Index (kg/m^2^)	23.1 (15.4–35.2)
Etiology (HBV/HCV/Alcohol/NAFLD or NASH	19/60/30/21
ALBI score (median (range))	−2.41 (−3.50–−1.55)
ALBI grade (1/2a/2b)	40/45/45
White blood cell (/µL)	4600 (1900–9800)
Neutrophil (%)	65 (34–86)
Lymphocyte (%)	23.5 (8–53)
AST (U/L)	41 (14–152)
ALT (U/L)	27 (14–179)
Tumor diameter (mm)	33 (10–136)
Number of tumors <5/≥5	39/91
BCLC stage (B/C)	69/61
Macrovascular invasion (No/Yes)	110/20
Extrahepatic spread (No/Yes)	84/46
AFP (ng/mL)	39.2 (1.2–284,543)
Treatment line (1st/2nd/3rd/4th)	72/46/8/4

Data are expressed as a median (range) or a number. Abbreviations: PS, performance status; HBV, hepatitis B virus; HCV, hepatitis C virus; NAFLD; nonalcoholic fatty liver disease, NASH; nonalcoholic steatohepatitis; ALBI, albumin–bilirubin grade; AST, aspartate transaminase; ALT, alanine aminotransferase; BCLC, Barcelona Clinic Liver Cancer; AFP, α-fetoprotein.

**Table 2 cancers-14-04284-t002:** Therapeutic responses according to RECIST (*n* = 130).

Initial Response	
CR	0 (0.0%)
PR	35 (26.9%)
SD	78 (60.0%)
PD	17 (13.1%)
ORR	35 (26.9%)
DCR	113 (86.9%)
**Best Response**	
CR	0 (0.0%)
PR	42 (32.3%)
SD	52 (40.0%)
PD	17 (13.1%)
ORR	42 (32.3%)
DCR	113 (86.9%)

Abbreviations: CR, complete response; PR, partial response; SD, stable disease; PD, progressive disease; ORR, objective response rate; DCR, disease control rate.

**Table 3 cancers-14-04284-t003:** Adverse events associated with Atezo/Beva (*n* = 130).

Adverse Event	Any *n* (%)	Grade 3 ≥ *n* (%)
Total adverse events	126 (96.9%)	47 (36.1%)
Liver injury	57 (43.8%)	8 (6.1%)
Hypertension	54 (41.5%)	9 (6.9%)
Proteinuria	37 (28.5%)	12 (9.2%)
Fatigue	36 (27.6%)	1 (0.7%)
Skin disorder	32 (24.6%)	1 (0.7%)
Fever	30 (23.0%)	2 (1.5%)
Hoarseness	21 (16%)	0 (0.0%)
Decreased appetite	20 (15.3%)	0 (0.0%)
Hypothyroidism	18 (13.8%)	0 (0.0%)
Diarrhea	17 (13.0%)	1 (0.7%)
Bleeding	17 (13.0%)	9 (6.9%)
Hypopituitarism	3 (2.3%)	3 (2.3%)
Heart failure	3 (2.3%)	3 (2.3%)
Drug-induced pneumonia	2 (1.5%)	2 (1.5%)
Infusion reaction	5 (3.8%)	0 (0.0%)

Abbreviations: Atezo/Beva, atezolizumab plus bevacizumab.

**Table 4 cancers-14-04284-t004:** Univariate and multivariate analyses of factors for OS.

	Univariate Analysis	Multivariate Analysis		
	*p*-Value	Odds Ratio	95% CI	*p*-Value
Age, <70 vs. ≥70	0.251			
Sex, male vs. female	0.246			
Etiology Viral, vs. non-viral	0.969			
m-ALBI grade, 1/2a vs. 2b	0.048	0.4773	0.239–0.953	0.042
BCLC, B vs. C	0.834			
AFP, <200 vs. ≥200 ng/mL	0.056			
Liver injury (Presence, vs. Absence)	0.036	2.400	1.201–4.798	0.016
Hypertension (Presence, vs. Absence)	0.001	0.311	0.134–0.720	0.006
Skin disorder (Presence, vs. Absence)	0.047	0.371	0.157–0.875	0.027
Post-progression treatment (Yes, vs. No)	0.005	0.271	0.134–0.549	0.005

Abbreviations: BCLC, Barcelona Clinic Liver Cancer; AFP, α-fetoprotein.

**Table 5 cancers-14-04284-t005:** Multivariate analysis factors associated with discontinuation due to AEs.

Factors	Unit	Odds Ratio	95% Confidence Interval	*p*
Fatigue grade ≥ 2	N/A	12.85	2.35–24.12	<0.001
Liver injury ≥ 3	N/A	6.29	1.54–19.33	<0.001
m-ALBI grade 2b	N/A	3.54	1.22–10.27	0.017

**Table 6 cancers-14-04284-t006:** Subsequent treatment after discontinuation of Atezo/Beva (*n* = 96).

Subsequent Treatment Rata	70.8% (68/96)
Lenvatinib	35.2% (24/68)
Ramucirumab	13.2% (9/68)
TACE	13.2% (9/68)
HAIC	8.8% (6/68)
Cabozantinib	7.4% (5/68)
Operation	7.4% (5/68)
Sorafenib	6.0% (4/68)
RFA	2.9% (2/68)
Radiation	1.4% (1/68)
others	4.5% (3/68)

Abbreviations: TACE; transcatheter arterial chemoembolization, HAIC; hepatic arterial infusion chemotherapy, RFA; radiofrequency ablation therapy.

## Data Availability

Data that support the findings of this study are available from the author, S.S. (Shigeo Shimose), upon reasonable request.

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
