# Peer review of "Association between Adverse Events and Prognosis in Patients with Hepatocellular Carcinoma Treated with Atezolizumab Plus Bevacizumab: A Multicenter Retrospective Study"

_cancers, 2022, doi:10.3390/cancers14174284_

Round 1
Reviewer 1 Report
The data you report are interesting in view of the need to personalize anti HCC therapy options and timely monitor treatment responses. Could you discuss in the aims of the study the prognostic role of AE also in terms of promptly switching to a second line therapy. However, I expect you to reply to and aknowledge my comments and request.
Author Response
Thank you very much for your letter regarding our manuscript (cancers-1851638). We appreciate your comments, which have helped us to improve our manuscript. In line with your comments, please find below our point-by-point responses.
- Could you discuss in the aims of the study the prognostic role of AE also in terms of promptly switching to a second line therapy. However, I expect you to reply to and aknowledge my comments and request.
Answer: Thank you for your valuable comment. Currently, sequential therapy with systemic drugs is considered effective for unresectable HCC [1, 2]. As you indicated, we should discuss the prognostic role of AE also in terms of promptly switching to second-line therapy.
In this study, the development of hypertension and skin disorders was a better OS. Kobayashi et al. reported that OS correlated with the treatment duration of systemic therapy [3]. In fact, we found that treatment duration was significantly longer in patients who developed these AEs than in patients who did not (median treatment duration: 7.8 vs. 4.9 months and 8.5vs. 5.3 months, p=0.001 and p=0.027, respectively). These AEs are relatively manageable using supportive drugs such as antihypertensive drugs and steroids. Therefore, as possible, it would be better to continue the treatment with the management for these AEs. However, if the effects of treatment are no longer confirmed, we should promptly switch to 2nd-line treatment. Moreover, treatment duration was significantly shorter in patients who developed liver injury than in patients who did not (median treatment duration: 4.3 vs. 6.7 months, p=0.034). Additionally, the median ALBI score in patients with liver injury did not improve compared with that at baseline, which suggests that the development of liver injury induces deterioration of liver function. Thus, AE management is needed for patients who developed liver injury as possible, but switching to second-line treatment should be also considered at the time of the development of liver injury, particularly severe liver injury. These descriptions regarding the correlation between AEs and treatment continuation were added in the revised manuscript (Line 295-309). Again, we appreciate your valuable comment, which has helped us to improve our manuscript
Reference
- Ogasawara, S., Y. Ooka, N. Itokawa, M. Inoue, S. Okabe, A. Seki, Y. Haga, M. Obu, M. Atsukawa, E. Itobayashi, H. Mizumoto, N. Sugiura, R. Azemoto, K. Kanayama, H. Kanzaki, S. Maruta, T. Maeda, Y. Kusakabe, M. Yokoyama, K. Kobayashi, S. Kiyono, M. Nakamura, T. Saito, E. Suzuki, S. Nakamoto, S. Yasui, A. Tawada, T. Chiba, M. Arai, T. Kanda, H. Maruyama, and N. Kato. "Sequential Therapy with Sorafenib and Regorafenib for Advanced Hepatocellular Carcinoma: A Multicenter Retrospective Study in Japan." Invest New Drugs 38, no. 1 (2020): 172-80.
- Tomonari, T., Y. Sato, J. Tani, A. Hirose, C. Ogawa, A. Morishita, H. Tanaka, T. Tanaka, T. Taniguchi, K. Okamoto, M. Sogabe, H. Miyamoto, N. Muguruma, K. Uchida, T. Masaki, and T. Takayama. "Comparison of Therapeutic Outcomes of Sorafenib and Lenvatinib as Primary Treatments for Hepatocellular Carcinoma with a Focus on Molecular-Targeted Agent Sequential Therapy: A Propensity Score-Matched Analysis." Hepatol Res 51, no. 4 (2021): 472-81.
- Kobayashi, K., S. Ogasawara, A. Takahashi, Y. Seko, H. Unozawa, R. Sato, S. Watanabe, M. Moriguchi, N. Morimoto, S. Tsuchiya, K. Iwai, M. Inoue, K. Ogawa, T. Ishino, T. Iwanaga, T. Sakuma, N. Fujita, H. Kanzaki, K. Koroki, M. Nakamura, N. Kanogawa, S. Kiyono, T. Kondo, T. Saito, R. Nakagawa, E. Suzuki, Y. Ooka, S. Nakamoto, A. Tawada, T. Chiba, M. Arai, T. Kanda, H. Maruyama, K. Nagashima, J. Kato, N. Isoda, T. Aramaki, Y. Itoh, and N. Kato. "Evolution of Survival Impact of Molecular Target Agents in Patients with Advanced Hepatocellular Carcinoma." Liver Cancer 11, no. 1 (2022): 48-60.

Reviewer 2 Report
Shimose et al. in their manuscript "Association between adverse events and prognosis in patients with hepatocellular carcinoma treated with atezolizumab plus bevacizumab: A multicenter retrospective study'' investigated the correlation between AEs and proggnosis for patients treated with atezolizumab plus bevacizumab. It is interesting to show that hypertension and skin disorders were favorable prognositic factors. This is an amazing study to explore the types of AEs may indicate different ouctome. While, there are several weakness should be noted.
To investigate the collinearity between hypertension, skin disorders and other covariates, multivariate cox regression should be performed, including factors of types AEs, post-progression treatment.
Furthermore, the correlation of AEs with response rate is an another interesting issue.
It may be interesting to analyze the early/late presence of AEs with survival outcome.
Author Response
Thank you very much for your letter regarding our manuscript (cancers-1851638). We appreciate your comments, which have helped us to improve our manuscript. In line with your comments, please find below our point-by-point responses.
1)To investigate the collinearity between hypertension, skin disorders and other covariates, multivariate cox regression should be performed, including factors of types AEs, post-progression treatment.
Answer: Thank you for your valuable comment. As you indicated, we should perform the analysis of the OS using multivariate cox regression, including factors of types AEs, and post-progression treatment.
In this analysis, liver injury (HR 2.400; p=0.016), hypertension (HR 0.311; P=0.006), skin disorder (HR 0.371; P=0.027), post-progression treatment (HR 0.271; P=0.005), and modified-ALBI 1 or 2a were identified for OS in multivariate cox regression. These descriptions were added to the revised manuscript (Revised Table 4, Line 201-209). Again, we appreciate your valuable comment, which has helped us to improve our manuscript.
Table 4. Univariate and multivariate analyses of factors for OS
|
|
Univariate analysis |
Multivariate analysis |
||
|
P-value |
Odds ratio |
95% CI |
P-value |
|
|
Age, <70 vs ≥70 |
0.251 |
|
|
|
|
Sex, male vs female |
0.246 |
|
|
|
|
Etiology Viral, vs non-viral |
0.969 |
|
|
|
|
m-ALBI grade, 1/2a vs 2b |
0.048 |
0.4773 |
0.239-0.953 |
0.042 |
|
BCLC, B vs C |
0.834 |
|
|
|
|
AFP, <200 vs. ≥200 ng/ml |
0.056 |
|
|
|
|
Liver injury (Presence, vs Absence) |
0.036 |
2.400 |
1.201-4.798 |
0.016 |
|
Hypertension (Presence, vs Absence) |
0.001 |
0.311 |
0.134-0.720 |
0.006 |
|
Skin disorder (Presence, vs Absence) |
0.047 |
0.371 |
0.157-0.875 |
0.027 |
|
Post-progression treatment (Yes, vs No) |
0.005 |
0.271 |
0.134-0.549 |
0.005 |
Abbreviations: BCLC, Barcelona Clinic Liver Cancer; AFP, α-fetoprotein
2)the correlation of AEs with response rate is an another interesting issue.
Answer: We appreciate your comment. Following your suggestion, we investigated the correlation between AEs and response rate. Although patients who developed hypertension had a significantly higher disease control rate (DCR) than those who did not, there were no significant differences in response rate between the development of other AEs with and without (Supplementally Table 2).
However, in terms of high DCR in patients who developed hypertension, these findings further support our studies that patients who developed hypertension survived significantly longer than those who did not. This issue was described in the revised manuscript (Table S2, Line 286-287).
Table S2. Therapeutic responses according to AEs (n=130)
|
|
All |
ORR (n=35) |
P |
DCR(n=114) |
P |
|
Liver injury (Presence/Absence) |
57/73 |
13/22 (22.8%/30.1%) |
0.349 |
48/66 (84.2%/90.4%) |
0.285 |
|
Hypertension (Presence/Absence) |
54/76 |
17/18 (31.4%/23.6%) |
0.323 |
53/61 (98.1%/80.2%) |
0.002 |
|
Proteinuria (Presence/Absence) |
37/93 |
9/26 (24.3%/27.9%) |
0.673 |
33/81 (89.1%/87.1%) |
0.743 |
|
Fatigue (Presence/Absence) |
36/94 |
14/21 (38.8%/22.3%) |
0.067 |
32/82 (88.8%/87.2%) |
0.797 |
|
Skin disorder (Presence/Absence) |
32/98 |
11/24 (34.4%/24.4%) |
0.273 |
30/84 (93.7%/85.7%) |
0.229 |
|
Fever (Presence/Absence) |
30/100 |
9/26 (30.0%/26.9%) |
0.664 |
25/89 (83.3%/89.0%) |
0.407 |
Abbreviations: RECIST, response evaluation criteria in solid tumors; ORR, objective response rate; DCR; disease control rate
3)It may be interesting to analyze the early/late presence of AEs with survival outcome.
Answer: We appreciate your comment. Following your suggestion, we analyze the early/late presence of AEs with survival outcomes. In this study, we defined the early presence of AEs that develop within 3 courses of treatment and the late presence of AEs that develop after 3 courses of treatment. There were no differences in OS between the two groups (Early-onset AEs:17.9 months vs. Late-onset AEs: not reached, P=0.658). This issue was described in the revised manuscript (Figure S3, Line 201-204).
